# Slope and distance from buildings are easy-to-retrieve proxies for estimating livestock site-use intensity in alpine summer pastures

**Marco Pittarello⬤\*, Simone Ravetto Enri⬤, Michele Lonati, Giampiero Lombardi⬤**

Department of Agricultural, Forest and Food Sciences, University of Torino, Grugliasco, Italy

\* marco.pittarello@unito.it

**Data Availability Statement:** We provide data supporting the finding of this study through an online repository (Zenodo), available at the

## Abstract

Regardless of the issue, most of the research carried out on summer pastures of European Alps had to consider the effects of grazing management, as it is an intrinsic component of alpine environment. The management intensity of grazing livestock is measured in terms of livestock stocking rate, but not always a direct measure of it is easily retrievable. Therefore, the aim of the research was to test the reliability of proxies easily retrievable from open data sources (i.e. slope and distance from buildings) in approximating the pastoral site-use intensity. To test the proxies' effectiveness two different approaches were used. With the first one, the proxies' reliability was assessed in a case-study conducted at farm scale by using the number of positions gathered with GPS collars, which are a reliable measure of livestock site-use intensity. With the second, the proxies' reliability was assessed by means of five Vegetation Ecological Groups (VEGs), used as a tool for indirect quantification of livestock site-use intensity at regional scale (thirty-two alpine valleys of the Western Italian Alps, Piedmont Region—Italy). At farm scale, distance from buildings and slope were both reliable predictors of the number of GPS locations as assessed with a Generalized Additive Model. Results of Generalized Linear Models at the regional scale showed that the values of both the slope and the distance from buildings were able to separate VEGs along the same site-use intensity gradient assessed by modelling the number of GPS locations at farm scale. By testing proxies' reliability both with a direct (i.e. GPS collar positions) and indirect (i.e. VEGs) measurement of livestock site-use intensity, results indicated that slope and distance from buildings can be considered effective surrogates of site-use intensity gradient in alpine grasslands managed under livestock grazing. Therefore, when the level of site-use intensity in research carried out in alpine summer pastures is not directly available, a reliable solution consists in the use of the terrain slope and the distance from buildings, which are also easily retrievable from open data sources or computable.

following web page through the DOI: 10.5281/zenodo.5578678.

**Funding:** Research was carried out in the framework of the "Pastoral vegetation types of Piedmont Alps" project (Principal Investigator prof. Andrea Cavallero) funded by Regione Piemonte. The funders had no role in study design, data collection and analysis, decision to publish, or preparation of the manuscript.

**Competing interests:** The authors have declared that no competing interests exist.

# Introduction

Up to date, research on summer pastures of the European Alps has been carried on extremely variegated topics, such as on soil microorganisms [1], plant diversity [2–5], animal diversity [6–8], nutrient cycle [9–11], forage quality [12, 13], forest dynamics [14], and many other issues. Regardless of the research topic, most of these studies considered the effects of grazing in their experimental design as it is an intrinsic component of alpine environment. Indeed, the agropastoral management, interacting with natural factors (e.g. climate, topography), has strongly shaped the Alps for millennials [15], influencing the ecological processes of all grassland natural resources (soil, plant and animal diversity, etc.). Since summer pastures are exploited by grazing livestock, rather than mown, the management intensity is often expressed in terms of animal stocking rate, i.e. the number of Livestock Units (LU) exploiting a given area over a specified period of time [16]. The grazing intensity of a specific area is well represented by the average stocking rate when such an area is relatively small and topographically homogeneous [17]. However, this condition is not common in alpine summer pastures, where slope, elevation, aspect, vegetation, etc. change frequently. Livestock prefer areas with gentle slopes, good accessibility, proximity to water sources and salt supplements, which generally exhibit a high forage quality and productivity [18, 19]. Vice-versa, the less accessible, less comfortable, farther away from water sources and salt supplements, and low-quality forage areas are generally discarded. As a consequence, this behavior results in an uneven spatial distribution of the stocking rate and of the site use intensity throughout the grazing area. However, livestock spatial distribution can be improved with an appropriate grazing management (e.g. rotational grazing, [20, 21]) and by the strategic placement of water troughs [22] and salt supplements [23].

The spatial-temporal distribution of livestock in alpine pastures can be measured or estimated for research or management purposes either directly or indirectly by using proper proxies. Direct measurements are available when the exploitation is planned, for instance, under an experimental design defining stocking rates a priori, i.e. setting up a specific number of LU over a known area and time-period. The usage of Global Positioning System (GPS) collars for livestock is another option, which allows to determine a detailed pattern of spatial use of a pasture and to punctually assess the grazing intensity [24]. The inconvenience of GPS collars is their high cost and for this reason they cannot be used in every research or applied to a large number of animals. Although some low-cost GPS tracking collars for livestock have been developed by readapting basic data loggers, they have a less reliable fix rate and fix schedule compared to commercial GPS [25]. Moreover, up to date, low-cost GPS have been tested only in USA [25–28] and Australian rangelands [29], and information about their performance is still unavailable in rough environments such as alpine pastures. As an alternative to GPS collars, animal positions can be mapped by direct observations [30]. Nonetheless, this approach has some limitations as well, such as the difficulty of recording positions during the night, the restricted duration of tracking (i.e. 1 or 2 days at time), the low location accuracy and the impact of the observers on livestock behavior [31]. Lastly, information regarding grazing activities and stocking rates can be gathered by farmers' interviews [32, 33].

When, due to a number of reasons, direct measurements are not applicable, the use of stocking rate proxies is often the only possibility, especially in large mountain pastures. For instance, the level of site use intensity exerted through grazing can be estimated from grassland botanical composition, as vegetation accurately reflects the level of nutrients redistributed by livestock excreta (feces and urine) and the level of forage intake pressure. Indeed, many authors found a strong relationship between the site use intensity gradient and the transition of specific vegetation communities [33, 34]. Tasser and Tappeiner [34] found a relation

between the gradient from nutrient-poor to nutrient-rich vegetation communities and the increase in land-use intensity. Likewise, Cavallero et al. [35] classified the vegetation of alpine summer pastures of Piedmont region (NW Italy) in grassland types (*sensu* Argenti and Lombardi, [36]), each one belonging to a Vegetation Ecological Group (hereafter 'VEG'). For instance, the shrub-encroached, oligotrophic, mesotrophic, eutrophic, and nitrophilous VEGs identified by Cavallero et al. [35] well characterize a gradient of increasing nutrient availability in the soil by means of livestock excreta, which in turn can be the expression of the pastoral site-use intensity. Vegetation-derived indexes are frequently used to estimate the site-use intensity as well. As mentioned above, pastoral management is one of the most important drivers of nutrients in the soil of alpine pastures since livestock take up nutrients by defoliation and return them through dung and urine [37]. Thereby the average indicator value for soil nutrient content (N) [*sensu* Landolt et al., 38] can be computed for any vegetation community to indirectly quantify the fertilization rate, and, therefore, the pastoral site-use intensity. Values can be either weighted with the abundance of each species found in a vegetation community [39] or not weighted [40, 41]. Nevertheless, as the determination of botanical composition is often time demanding, several other site-use intensity proxies have been experienced. For example, Dorji et al. [42] and Ravetto Enri et al [22] used distance from congregation areas and water sources. However, the distance from available water sources could be difficult to determine in some circumstances, such as when only natural streams are available over the grazing area. These water sources often provide water availability seasonally or within a limited period. On the other hand, a topographic variable such as slope is strongly related to site-use intensity as steep slopes can represents a limit both for human activities [2, 43] and for livestock exploitation [18]. Elevation also is an important topographic variable related to site-use intensity, especially when considering a wide elevation range (i.e. from lowland to high elevation pastures) rather than to a narrow elevation range (i.e. within a single pasture). In addition, also the distance from buildings (e.g. farm center, cowshed, etc.) is often adopted [34, 44–46]. Generally, the greater the distance from such buildings, the lower is the site-use intensity.

For sure, indirect measures of site-use intensity are less precise than direct measurements, but they often are the only feasible option. Slope and distance from buildings are amongst the easiest proxies of site-use intensity to retrieve. Slope can be computed in a Geographic Information System (GIS) environment by using a Digital Terrain Model (DTM) downloadable from regional and national government geo-portals or from European [e.g. European Environment Agency, 47] and worldwide DTM repositories (e.g. ASTER Data, [48]). Building positions can be extracted for free by specific web services, such as BBBike [49], which allow to extract several layers from Planet.osm (i.e. OpenStreetMap data) in shapefile format.

Since to our knowledge no previous studies have analytically assessed the reliability of easy-to-retrieve proxies in estimating pastoral site-use intensity, the aim of the research was to test the effectiveness of slope and distance from buildings in approximating the pastoral site-use intensity of alpine pastures. To achieve such aim, two different approaches were used. With the first one, the reliability of the selected proxies was assessed in a case-study conducted at farm scale by using the number of positions gathered with GPS collars, which are a precise tool for quantifying livestock site-use intensity. With the second, the reliability of the selected proxies was assessed by means of the botanical composition, used as a tool for indirect quantification of livestock site-use intensity. Once ascertained that vegetation could describe a gradient of site-use intensity, the goodness of slope and distance from buildings in discriminating VEGs related to a pastoral site-use intensity gradient was assessed at a wide territorial scale (regional) using a dataset of vegetation surveys carried out across the pastures of Western Italian Alps.

## Materials and methods

### Study areas

**Farm scale.**   The research at farm scale was conducted in Val Troncea Natural Park, Piedmont region, in the south-western Italian Alps (lat. 44˚570N, long. 6˚570E). Fieldworks and the animal experimental usage were authorized by Val Troncea Natural Park management authority and the farmer, respectively. The study area was characterized by three large pastures with a mixture of grasslands and open shrublands, located between 1900 and 2800 m a.s.l. (Table 1 and Fig 1). Grasslands were dominated by *Festuca curvula* Gaudin, *Festuca violacea* aggr., *Festuca quadriflora* Honk, *Helianthemum nummularium* L., *Festuca rubra* aggr. and *Agrostis tenuis* Sibth. The shrub layer was mainly dominated by *Vaccinium gaultherioides* Bigelow, *Juniperus nana* Wild, *Rhododendron ferrugineum* L. and *Vaccinium myrtillus* L. Soils, originated from calcareous bedrock, were gravelly and nutrient poor. For the period 2003–2015, the average annual temperature is about 4˚C (Feb: −3.8˚C; Jul: 12.6˚C) and annual average precipitation is 703 mm (data from meteorological station located at 2150 m a.s.l.– 44˚98'N, 6˚94'E). The three pastures were managed with a rotational grazing system from June 23 to August 30, 2014. Pastures were grazed following the seasonal patterns of forage growth, i.e. from the one at lowest altitude to the one at the highest altitude, by a herd of 87 LU of Piedmontese beef cows. The herd included heifers, non-lactating cows and suckler cows with an age ranging from 1 to 15 years.

**Regional scale.**   The study at regional scale was carried out across the summer pastures of 32 alpine valleys of the Western Italian Alps, Piedmont Region (Italy), from the mountain to the alpine belts (490 to 2900 m a.s.l., Fig 1). Mean total annual precipitation ranged between 640 mm and 1600 mm in the western and northern sectors of the region, respectively (data 1977–2007, [50]). The predominant massifs are Crystalline, which also alternate with calcareous rocks of sedimentary origin (limestones, dolomites, schists, etc.) [51]. Summer pastures occurred over a wide range of nutrient, elevation, and water soil availability (see Pittarello et al., [52] for details).

### Global positioning system tracking

In agreement with the farmer, 14 cows were randomly selected for being tracked with GPS collars at farm scale (GPS Model Corzo, Microsensory SLL Fernàn Nùñez, Andalusia, Spain). In rugged mountain terrain, these collars have an average accuracy of 5 m. Ten days before the start of the grazing season, the collars were placed to the animals so that they could get used to them. Position recording was set at 15-minute intervals to ensure battery life for the entire grazing season.

### Vegetation data

The area was subdivided into 25 x 25-m grid cells for the farm scale study, resulting in a grid with 2339 cells, each used as a sample unit. Vegetation types were attributed to each grid cell

**Table 1. Characteristics of the three pastures located at Val Troncea Natural Park (Italy) and used for the farm-scale study.**

| Pasture | Area | Grassland cover | Grazable area | Average elevation | Average slope |
|---|---|---|---|---|---|
| | (ha) | (%) | (ha) | (m asl) | (˚) |
| 1 | 26.6 | 71 | 19.1 | 2005 | 25.2 |
| 2 | 45.2 | 61 | 27.8 | 2278 | 26.8 |
| 3 | 59.7 | 54 | 32.2 | 2598 | 25.2 |

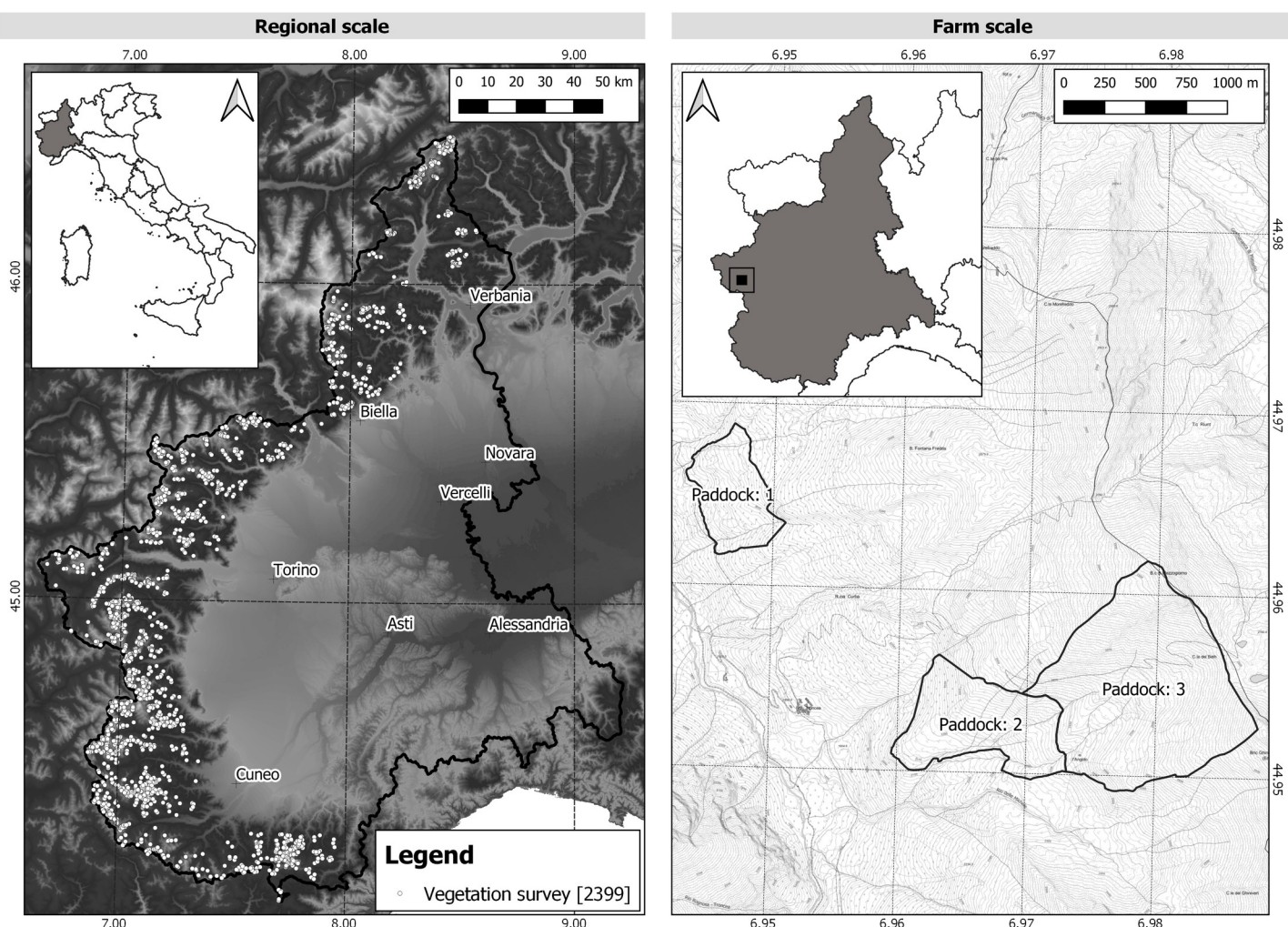

**Fig 1. Distribution of 2399 vegetation surveys belonging to shrub-encroached, oligotrophic, mesotrophic, eutrophic, and nitrophilous Vegetation Ecological Groups spread over the summer pastures of Piedmont Alps and used for the regional-scale study (Image on the left).** Location of the three large pastures in Val Troncea Natural Park, Piedmont region, used for the farm-scale study (Image on the right). Coordinate reference system: WGS84. Background map sources: Copernicus Land Monitoring Service—EU-DEM of the European Environment Agency (Image on the left) and maps drawn by Disafa based on Cartographic Reference Layout BDTRE 2018 Black/White 1:10.000 of the Piedmont Region. (Image on the right).

based on a vegetation map drawn in a previous research [53]. Vegetation types are communities occurring in similar ecological conditions, dominated by 1–2 (3) species together with a constant occurrence of a variable number of common species [35].

At regional scale, vegetation data were retrieved from the dataset of Cavallero et al. [35]. This dataset counts 3888 vegetation surveys—used as sample units—carried out in the period 2001–2007, spread over about 60.000 ha of summer pastures in Piedmont and classified into 92 different vegetation types. The position of each survey was recorded with a hand-held GPS.

Only the sample units belonging to shrub-encroached, oligotrophic, mesotrophic, eutrophic, and nitrophilous VEGs as defined by Cavallero et al. [35] were retained for further analyses since they occurred both at farm and regional scales (S1 Table). These communities as sorted according to a soil nutrient gradient were considered the result of an increasing gradient of site-use intensity. The number of the retained vegetation sample units and vegetation surveys used in subsequent analyses were 1145 (49.0%) and 2399 (61.7%) of the farm- and the

**Table 2. Count and relative percentage of the sample units and vegetation surveys retained for the five Vegetation Ecological Groups (VEG) at farm and regional scale, respectively.**

| VEG | Farm scale | | Regional scale | |
|---|---|---|---|---|
| | number | % | number | % |
| Shrub-encroached | 236 | 20.60% | 49 | 2.00% |
| Oligotrophic | 475 | 41.50% | 1178 | 49.10% |
| Mesotrophic | 372 | 32.50% | 875 | 36.50% |
| Eutrophic | 58 | 5.10% | 271 | 11.30% |
| Nitrophilous | 4 | 0.30% | 26 | 1.10% |
| | 1145 | 100.00% | 2399 | 100.00% |

regional-scale databases, respectively. The proportion of sample units classified in each VEGs at the two spatial scales is reported in Table 2.

## Computation of site-use intensity proxies

At farm scale, the slope was calculated as the average value of each 25-m grid cell from a 25-m resolution DTM [54]. At regional scale, the slope was determined by spatially intersecting the vegetation survey positions and the same DTM used at farm scale.

To calculate the distance from cowsheds, farm centers and other buildings related to pastoral activities (foothold for pastoral equipment such as mobile electric fences, posts, energizers), the building positions of the whole regional alpine chain were extracted by using BBBike [49]. Then, the distance from each grid cell centroid and from each vegetation survey to the nearest building was computed for the farm and regional study. Proximity was based on a straight-line distance between vegetation surveys and buildings. All spatial analyses were carried out in GIS environment [55, 56].

## Data analyses

To assess if slope and distance from buildings were good predictors of site-use intensity at farm level, the number of GPS locations counted within sample units was modelled as a function of the two proxies. Since the scatterplots of the response variable versus the two proxies did not show a clear linear pattern (Fig 2), a Generalized Additive Model (GAM) was preferred instead of a Generalized Linear Model (GLM). The additive models fit smoothing curves through the data and for this reason they are preferable when the relationships amongst the response variable and covariates do not show a strict linear pattern [57]. Since the slope and the distance from buildings were not correlated (Pearson $r = 0.0067$; $p = 0.82$), they were both retained as covariates for modelling the number of GPS locations counted in the sample units. The smoother basis set for both the covariates was a "thin plate regression spline" and the number of knots was set to 10 and 15 for slope and distance from buildings, respectively. Such numbers of knots were chosen as they ensured a number of degrees of freedom sufficient for an adequate smoothing [58]. The number of GPS locations was spatially autocorrelated amongst sample units (Moran's I, $p < 0.001$), so that the spatial effects were incorporated in the GAM by fitting a smoothed term with an interaction between latitude and longitude of sample unit centroids and with a Gaussian process basis (using a power exponential covariance function). Total number of GPS locations, being a count variable, was modelled with both Poisson and Negative Binomial distributions. Then, the model resulting in the lowest Akaike's Information Criterion (AIC) value was considered as the best fitting one [57]. Model assumptions, i.e. residual normality, heteroscedasticity, independence, and response vs fitted values,

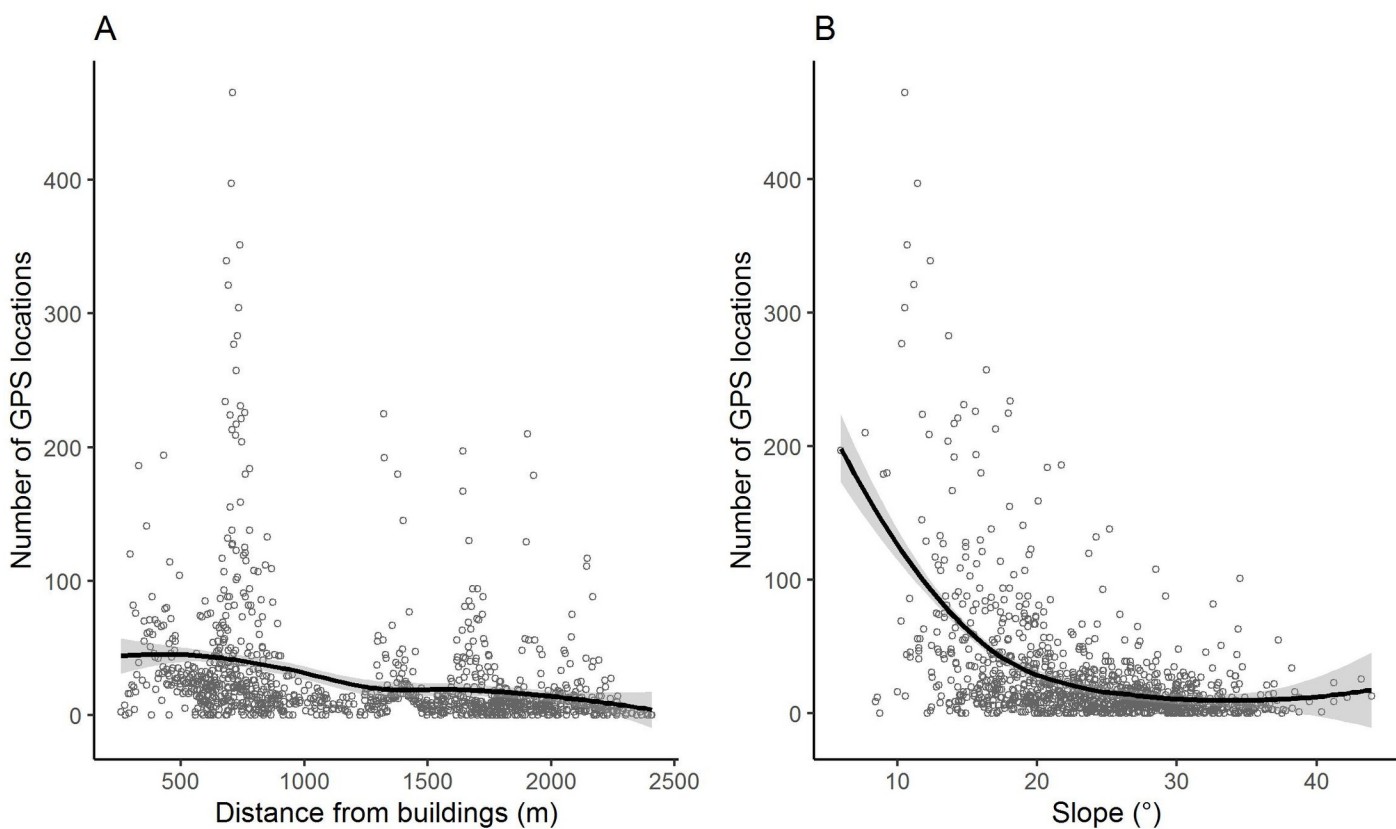

**Fig 2.** Scatterplots of the number of GPS locations versus the distance from buildings (A) and slope (B). A local weighted regression smoothing (LOESS) curve was added in each panel.

were graphically checked using Pearson residuals [57] (S1 Fig). Residual spatial autocorrelation was assessed using Moran's I [59]. By including in the model the smoothed term of the interaction between latitude and longitude of sample unit centroids, the spatial autocorrelation was no longer detected (S1 Fig).

A two-step analysis was carried out to test site-use intensity proxies with VEGs. A preliminary correspondence between measured pastoral site-use intensity and VEGs was assessed at the farm scale. To evaluate if the expected transition of VEGs from the shrub-encroached to the nitrophilous ones corresponded to a real site-use intensity gradient as represented by the stocking rates measured through GPS locations, the differences among the five VEGs in terms of the total GPS locations were assessed. A GLM was used by specifying the total number of GPS locations within sample units as response variable and the VEG as fixed factor. Tukey's post-hoc tests with Bonferroni-adjusted $p$-values were performed to outline significant differences among VEGs. Being the total number of GPS locations a count variable, it was modelled with both Poisson and Negative Binomial distributions. Similarly to previous GAM analysis, the model resulting in the lowest Akaike's Information Criterion (AIC) value was retained. At a second stage, the proxies were tested with VEGs at regional scale. To evaluate if the five VEGs were effectively discriminated using proxies, two GLMs were performed by specifying either slope and distance from buildings as response variables and VEG as fixed factor. The tests were followed by Tukey's post-hocs with Bonferroni adjustment to examine the differences among single VEGs. Being slope and distance from buildings continuous variables, they were modelled with a normal distribution.

**Table 3. Summary of the Generalized Additive Model (GAM) used to assess the effects of slope and distance from buildings on the number of GPS locations, counted within sample units at farm scale.**

| Smoothed term | edf | Ref.df | Chi.sq | P-value | P |
|---|---|---|---|---|---|
| s(Slope) | 3.72 | 4.72 | 129.72 | <2e-16 | *** |
| s(Distance from buildings) | 9.85 | 10.28 | 18.17 | 0.0383 | * |
| s(Latitude·longitude) | 97.16 | 106.22 | 1728.63 | <2e-16 | *** |

's(latitude·longitude)' indicates the smoothed interaction term between latitude and longitude of sample unit centroids. EDF is the effective degree of freedom and Ref. DF is the reference degrees of freedom used in the statistical test of "no effect" of each independent variable. Chi-sq is the test statistic.

Family distribution: Negative binomial (log link).

Adjusted $R^2$: 0.844.

Deviance explained: 83.8%.

The R software [60] was used for statistical analyses. The GAM was performed with the 'mgcv' package [58]. GLMs and Tukey's post-hocs were performed with the 'glmmTMB' [61] and 'emmeans' packages respectively [62]. Moran's I tests were performed with the 'ape' package [63].

## Results

At farm level, both the slope and the distance from buildings were significantly and negatively related to the number of GPS locations, according to the GAMs (Table 3 and Fig 3). The

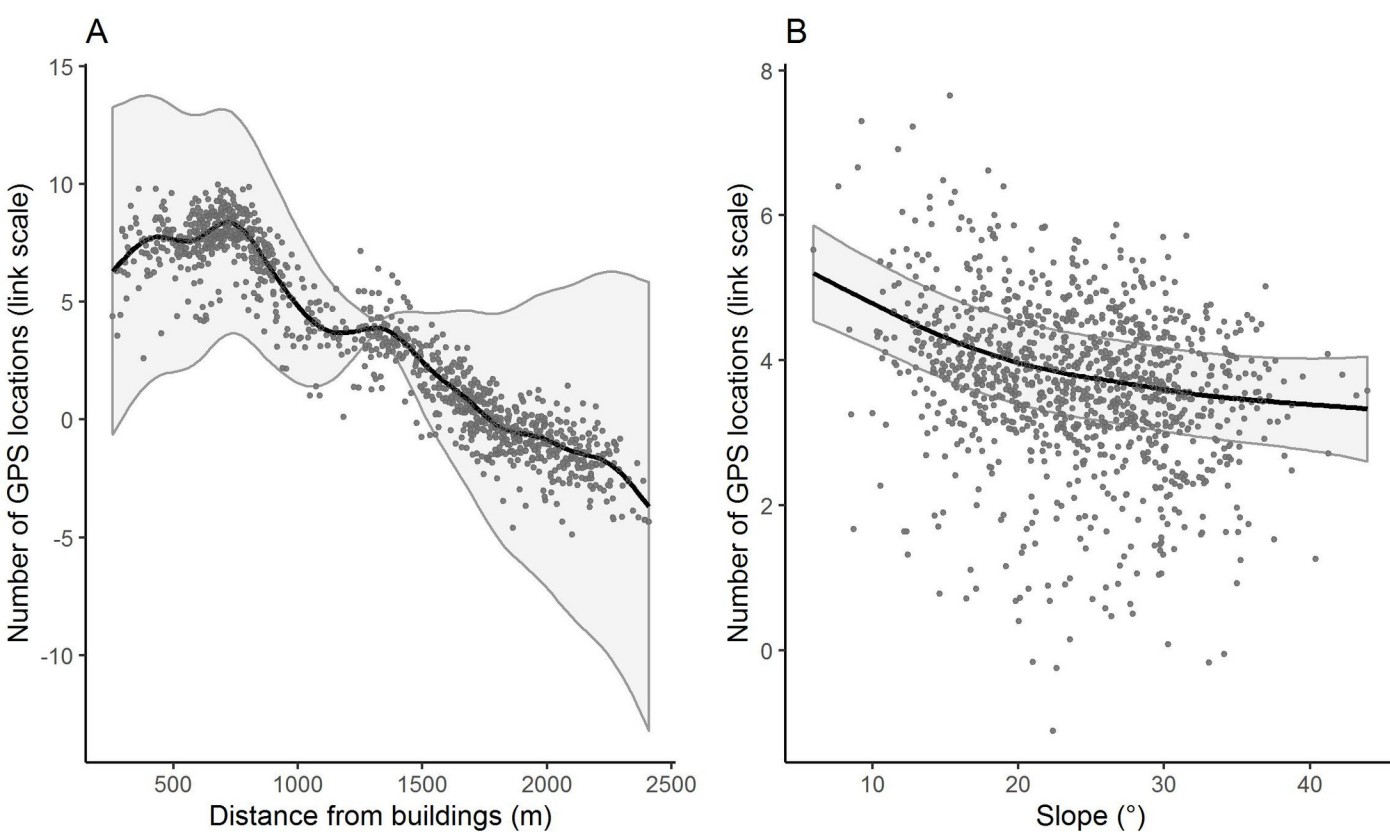

**Fig 3.** Estimated smoothing curves for the distance from buildings (A) and the slope (B) from the Generalized Additive Model (GAM). The response variable is the number of GPS locations in the link scale unit of the GAM.

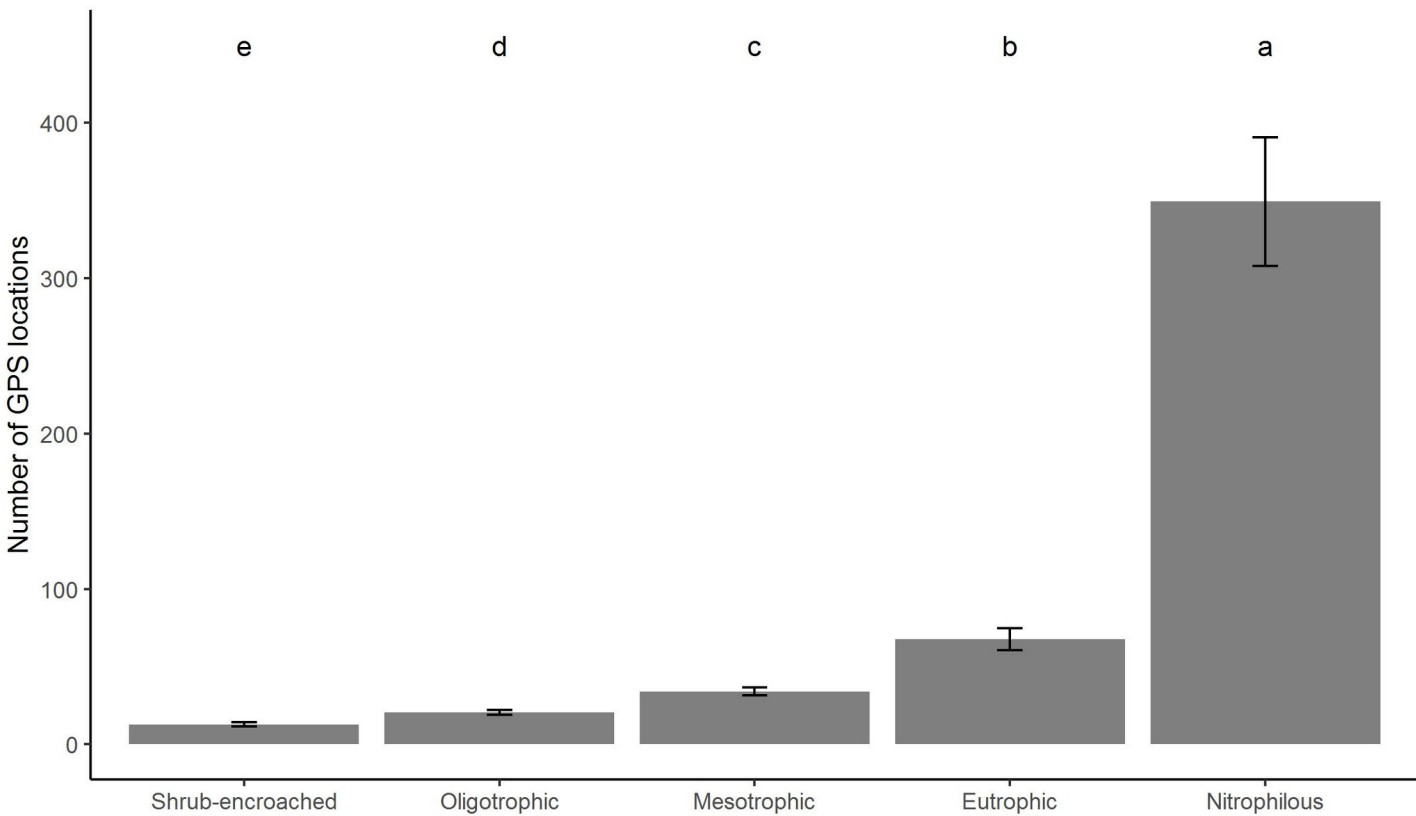

**Fig 4. Mean number of GPS locations for each Vegetation Ecological Group at the farm-scale study.** Error bars represent the standard error of the mean, whereas letters above histograms indicate significant differences according to Tukey's post-hoc test with Bonferroni-adjusted *p*-values.

estimated degrees of freedom of both the covariates were higher than 1, meaning that the smoothing splines were far from representing a linear relationship. The model showed an adjusted $R^2$ of 0.844 and the deviance explained was 83.8%.

The number of GPS locations at farm scale differed amongst VEGs (F ratio = 46.503, P<0.001). More specifically, the number of GPS locations increased significantly from shrub-encroached to nitrophilous VEGs (Fig 4).

Results of GLMs at the regional scale showed that the values of both the slope and the distance from buildings differed among the five VEGs (slope: F ratio = 22.127, P < 0.001; distance from buildings: F ratio = 23.978, P < 0.001). The two variables were able to separate VEGs along the same site-use intensity gradient assessed by modelling the number of GPS locations (Fig 5) at farm scale. Shrub-encroached areas were placed in the steepest sites, followed by oligo- and mesotrophic VEGs and then by eutrophic ones, which occurred on the gentler slopes. Nitrophilous VEGs were placed in sites with slopes comparable to those of oligo-, meso-, and eutrophic ones. Shrub-encroached and oligotrophic VEGs occurred farther away from buildings, whereas the eutrophic and nitrophilous ones closer to buildings. Mesotrophic VEGs were between nitrophilous-eutrophic and shrub-encroached ones in terms of distance from buildings.

## Discussion

The GAMs at farm scale indicated that distance from buildings and slope were both reliable predictors of the site-use intensity in alpine pastures. According to the estimated smoothing

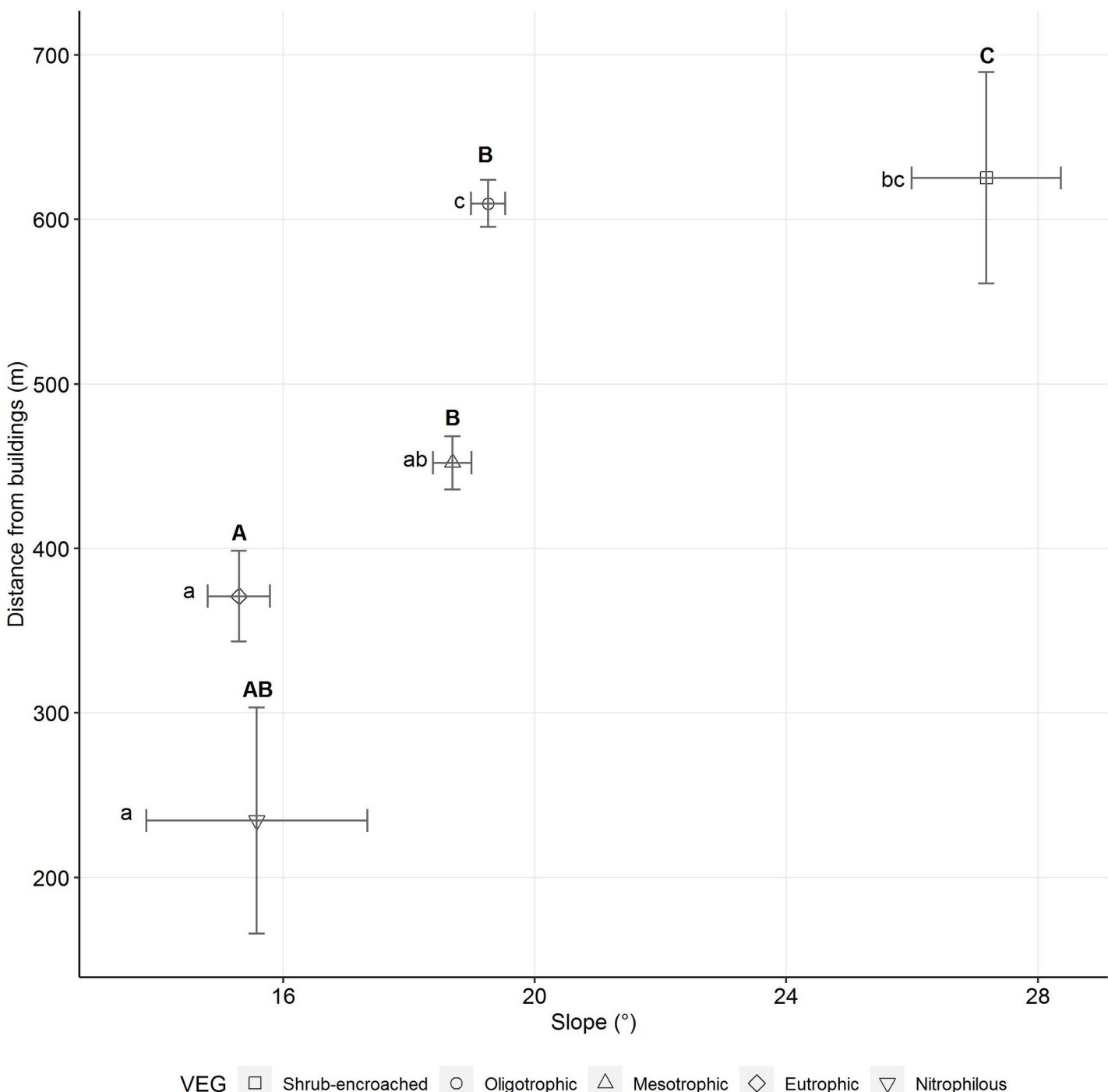

**Fig 5. Mean values of slope and distance from buildings for each Vegetation Ecological Group at regional-scale study.** Horizontal and vertical error bars represent the standard error of the mean for the slope and the distance from buildings, respectively. Capital and lower-case letters indicate significant differences according to Tukey's post-hoc test with Bonferroni-adjusted p-values for the slope and the distance from buildings, respectively.

curves of the GAM, the steeper and the farther from buildings were, the lower the number of GPS locations was. Considering that site-use intensity was measured by means of GPS collars, which are considered one of the most accurate tools for studying livestock spatial patterns and

an accurate measurement of the site-use intensity [31], this result can be considered of a note-worthy value. Moreover, the GAM's adjusted $R^2$ and the deviance explained were both high, suggesting a good model fit and that the proxies effectively explained a high portion of the variance of the number of GPS locations.

Even the indirect quantification of livestock site-use intensity by means of VEGs, provided results in line with those of the GAMs approach. Firstly, the study at farm scale confirmed the strict connection between the stocking rate, as assessed with GPS collars, and VEGs. Through the GPS fixes gathered in this study, a site-use intensity gradient was clearly identified and linked to a transition of specific vegetation communities along a gradient of nutrient availability. Indeed, plant communities in alpine pastures generally show a patchy distribution, as a result of the mutual interaction between biotic (livestock grazing) and abiotic (climate, terrain morphology and slope, bedrock type, soil properties, aspect, etc.) factors [52]. The combined effects of several pastoral management factors (trampling, grazing, seed transportation, nutrient redistribution due to dung deposition, and feeding behavior) heavily affect vegetation botanical composition [64], which in turn can mirror the level of site-use intensity. Secondly, being VEGs an accurate surrogate of the number of GPS locations, they were used for assessing differences in terms of slope and distance from buildings at a larger scale. Distance from buildings and slope allowed to properly order the five VEGs at regional scale along the same site-use intensity gradient defined by the number of GPS locations at farm level. The distance from buildings described a spatial gradient from the edifices, where livestock likely concentrate for sheltering and milking, to the surrounding vegetation, since the higher the distance, the lower was the forage quality and productivity because of a lower input of nutrients distributed with livestock excreta. Tasser and Tappeiner [34] demonstrated that vegetation communities farther away from cowsheds were characterized by a lower percentage of grazing-tolerant plant species and by a higher percentage of dwarf shrubs, symptomatic of a long-lasting undergrazing condition. The distance from buildings should be a proxy more related to dairy farming systems, where animals graze during the day and then they are gathered at the farm center during the night for milking and resting, rather than beef farming systems where animals are typically not sheltered at night in such structures. However, even though the farm-scale study was carried out in a beef farm, the distance from buildings properly identified a transition of VEGs as well. Indeed, buildings are useful as a foothold for pastoral equipment (e.g. mobile electric fences, posts, energizers), protection against predators, and veterinary controls.

Concerning slope, gentler terrains are suitable places for livestock to rest and ruminate. Since livestock spend a large amount of time in such comfortable places, an enrichment of nutrients in the soil generally occurs due to a high dung and urine supply, which, in the worse cases, could lead to nitrophilous vegetation dominated, for instance, by *Rumex alpinus* L., *Chenopodium bonus-henricus* L. and *Urtica dioica* L.). Conversely, steep slopes are uncomfortable sites for livestock and, being scarcely exploited, the level of supplied nutrient is low [65] and the vegetation dominated by dwarf shrubs (e.g. *Rhododendron ferrugineum* L. and *Juniperus nana* Willd.) and oligotrophic species (e.g. *Nardus stricta* L., *Carex sempervirens* Vill., and *Carex curvula* All.). Nitrophilous VEGs were the hardest to disentangle from the others through the proxies, as their values did not statistically differ from those of oligo-, meso-, and eutrophic ones. However, the low number of nitrophilous vegetation samples may have lessened the effective discrimination from the other VEGs. The slope and the distance from buildings did not allow the identification of an exact value of site-use intensity, but they were able to adequately describe its gradient across different types of vegetation communities.

By testing proxies' reliability both with a direct (i.e. GPS collar positions) and indirect (i.e. botanical composition) measurement of livestock site-use intensity, results indicated that slope

and distance from buildings can be considered effective surrogates of site-use intensity gradient in alpine grasslands managed under livestock grazing.

## Conclusions

When research carried out in alpine summer pastures have to consider the level of site-use intensity, which is generally strictly mirrored by livestock stocking rate, measuring directly the variables that explain a specific phenomenon is not always possible. This study demonstrated that a reliable solution consists in the use of the terrain slope and the distance from buildings used for the pastoral management, which, besides, are easily retrievable from open data sources or computable.

## Supporting information

**S1 Fig. Validation plots for the Generalized Additive Model (GAM).**
(TIF)

**S1 Table. Ecological scheme proposed by Cavallero et al.** (2007) in which vegetation types are grouped in vegetation ecological groups, i.e. vegetation communities with similar ecological needs. Phytosociological plant communities are indicated for each vegetation type according to Grabherr and Mucina [66] and Mucina et al. [67, 68].
(XLSX)

## Acknowledgments

The authors are grateful to the Giletta Farm for supplying the livestock and for collaborating on the farm scale study, and to Val Troncea Natural Park for its continued assistance. Special thanks are addressed to all the people who helped with the fieldwork.

## Author Contributions

**Conceptualization:** Marco Pittarello, Simone Ravetto Enri, Michele Lonati.

**Data curation:** Marco Pittarello.

**Formal analysis:** Marco Pittarello.

**Funding acquisition:** Giampiero Lombardi.

**Methodology:** Marco Pittarello.

**Project administration:** Giampiero Lombardi.

**Validation:** Marco Pittarello.

**Visualization:** Marco Pittarello.

**Writing – original draft:** Marco Pittarello.

**Writing – review & editing:** Marco Pittarello, Simone Ravetto Enri, Michele Lonati, Giampiero Lombardi.

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
