## [Decision Letter · Decision Letter 0]

28 Jul 2021

PONE-D-21-19475

Slope and distance from buildings are easy-to-retrieve proxies for estimating livestock site-use intensity in alpine summer pastures

PLOS ONE

Dear Dr. Pittarello,

Thank you for submitting your manuscript to PLOS ONE. After careful consideration, we feel that it has merit but does not fully meet PLOS ONE’s publication criteria as it currently stands. Therefore, we invite you to submit a revised version of the manuscript that addresses the points raised during the review process. As you will see, our reviewer has only pointed some minor metodological details that should be clarified in a revised version.

We look forward to receiving your revised manuscript.

Kind regards,

Emmanuel Serrano, PhD

Academic Editor

PLOS ONE

3. In your Methods section, please provide additional details regarding participant consent from the owners of the animals. In the ethics statement in the Methods and online submission information, please ensure that you have specified (1) whether consent was informed and (2) what type you obtained (for instance, written or verbal). If the need for consent was waived by the ethics committee, please include this information.

4. We note that you have stated that you will provide repository information for your data at acceptance. Should your manuscript be accepted for publication, we will hold it until you provide the relevant accession numbers or DOIs necessary to access your data. If you wish to make changes to your Data Availability statement, please describe these changes in your cover letter and we will update your Data Availability statement to reflect the information you provide

5. We note that Figure 1 in your submission contain map images which may be copyrighted. All PLOS content is published under the Creative Commons Attribution License (CC BY 4.0), which means that the manuscript, images, and Supporting Information files will be freely available online, and any third party is permitted to access, download, copy, distribute, and use these materials in any way, even commercially, with proper attribution. For these reasons, we cannot publish previously copyrighted maps or satellite images created using proprietary data, such as Google software (Google Maps, Street View, and Earth). For more information, see our copyright guidelines: http://journals.plos.org/plosone/s/licenses-and-copyright.

Reviewers' comments:

Reviewer's Responses to Questions

**Comments to the Author**

1. Is the manuscript technically sound, and do the data support the conclusions?

Reviewer #1: Yes

2. Has the statistical analysis been performed appropriately and rigorously? 

Reviewer #1: I Don't Know

3. Have the authors made all data underlying the findings in their manuscript fully available?

Reviewer #1: Yes

4. Is the manuscript presented in an intelligible fashion and written in standard English?

Reviewer #1: Yes

5. Review Comments to the Author

Reviewer #1: General assessment:

The paper provides interesting information in the field of grazing pressure in alpine pastures. The subject falls within the general scope of the PLOSONE journal. The originality of the work’s contribution lies in demonstrating that slope and distance to buildings along with the types of vegetation and the use of GPS collars contribute to explain the use of grasslands. The methodology is adequate. The interpretations are well justified and consistent with the objectives.

Listed below are several minor aspects that can clarify the article. With attention to these points the paper should be acceptable.

Specific points:

- Remove double parentheses in lines 54, 82, 90, 110, 111 and 146.

- Line 66. There are some other works that used low-cost GPS collars (e.g. Hanpson et al 2010. Monitoring distances travelled by horses using GPS tracking collars. Australian Veterinary Journal).

- Lines 74-77. Authors must provide a bibliographic citation that accredits this sentence.

- Lines 103-104. Authors should detail the types of buildings considered.

- Lines 120-121. This sentence requires a citation too.

- Line 182. Clarify if the distance considered was in a straight line or following the available paths.

- Line 187. Clarify if for the calculation of the stocking rate only the animal that carried the GPS or the entire herd was considered.

6. PLOS authors have the option to publish the peer review history of their article (what does this mean?). If published, this will include your full peer review and any attached files.

Reviewer #1: **Yes: **Jordi Bartolomé Filella

---

## [Author Response · Author response to Decision Letter 0]

3 Sep 2021

PONE-D-21-19475

Slope and distance from buildings are easy-to-retrieve proxies for estimating livestock site-use intensity in alpine summer pastures

PLOS ONE

ANSWER: We modified the manuscript style accordingly 

ANSWER: We specified in Methods section that all the permissions and permits required for fieldwork and utilization of cattle were requested and authorized by the administration of Val Troncea Natural Park.

3. In your Methods section, please provide additional details regarding participant consent from the owners of the animals. In the ethics statement in the Methods and online submission information, please ensure that you have specified (1) whether consent was informed and (2) what type you obtained (for instance, written or verbal). If the need for consent was waived by the ethics committee, please include this information.

ANSWER: We specified in Methods section that in agreement with the farmer 14 cows were randomly selected for being tracked with GPS collars. 

4. We note that you have stated that you will provide repository information for your data at acceptance. Should your manuscript be accepted for publication, we will hold it until you provide the relevant accession numbers or DOIs necessary to access your data. If you wish to make changes to your Data Availability statement, please describe these changes in your cover letter and we will update your Data Availability statement to reflect the information you provide

ANSWER: We specified in the submission form, at the “Additional information” section, at the point “Describe where the data may be found in full sentences. If you are copying our sample text, replace any instances of XXX with the appropriate details.” that the data that support the findings of this study are available from the corresponding author upon reasonable request.

5. We note that Figure 1 in your submission contain map images which may be copyrighted. All PLOS content is published under the Creative Commons Attribution License (CC BY 4.0), which means that the manuscript, images, and Supporting Information files will be freely available online, and any third party is permitted to access, download, copy, distribute, and use these materials in any way, even commercially, with proper attribution. For these reasons, we cannot publish previously copyrighted maps or satellite images created using proprietary data, such as Google software (Google Maps, Street View, and Earth). For more information, see our copyright guidelines: http://journals.plos.org/plosone/s/licenses-and-copyright.

ANSWER: The image on the left has the digital surface model (DSM) of the European Environment Agency (EEA) as background. At the following link https://www.eea.europa.eu/data-and-maps/data/copernicus-land-monitoring-service-eu-dem at the point of Rights is reported: “Access to data is based on a principle of full, open and free access as established by the Copernicus data and information policy Regulation (EU) No 1159/2013 of 12 July 2013. This regulation establishes registration and licensing conditions for GMES/Copernicus users. Free, full and open access to this data set is made on the conditions that:

1.When distributing or communicating Copernicus dedicated data and Copernicus service information to the public, users shall inform the public of the source of that data and information.”

The image on the right is a map available from the repository of the Piemonte Region (http://www.geoportale.piemonte.it/geocatalogorp/?sezione=catalogo) and it is covered by the Creative Commons license (see http://www.datigeo-piem-download.it/direct/Geoportale/RegionePiemonte/Licenze/New/Licenza_CC40BY.pdf)

According to these indications we have reported in the legend of Figure 1 the sources of the images used as backgrounds.

ANSWER: We do not have any retracted article

 

Reviewers' comments:

Reviewer's Responses to Questions

Comments to the Author

1. Is the manuscript technically sound, and do the data support the conclusions?

Reviewer #1: Yes

2. Has the statistical analysis been performed appropriately and rigorously?

Reviewer #1: I Don't Know

3. Have the authors made all data underlying the findings in their manuscript fully available?

Reviewer #1: Yes

4. Is the manuscript presented in an intelligible fashion and written in standard English?

Reviewer #1: Yes

5. Review Comments to the Author

Reviewer #1: General assessment:

The paper provides interesting information in the field of grazing pressure in alpine pastures. The subject falls within the general scope of the PLOSONE journal. The originality of the work’s contribution lies in demonstrating that slope and distance to buildings along with the types of vegetation and the use of GPS collars contribute to explain the use of grasslands. The methodology is adequate. The interpretations are well justified and consistent with the objectives.

Listed below are several minor aspects that can clarify the article. With attention to these points the paper should be acceptable.

Specific points:

- Remove double parentheses in lines 54, 82, 90, 110, 111 and 146.

ANSWER: We modified the text accordingly

- Line 66. There are some other works that used low-cost GPS collars (e.g. Hanpson et al 2010. Monitoring distances travelled by horses using GPS tracking collars. Australian Veterinary Journal).

ANSWER: Thank you! We added the reference you suggested and also others related to the use of low-cost GPS collars (26 – 29). These research have been carried out in USA and Australian rangelands. Therefore, we modified the text accordingly: “Moreover, up to date, low-cost GPS have been tested only in USA [25-28] and Australian rangelands [29]) and information about their performance is still unavailable in rough environments such as alpine pastures.” 

26. Brennan, J., Johnson, P., & Olson, K. Classifying season long livestock grazing behavior with the use of a low-cost GPS and accelerometer. Computers and Electronics in Agriculture, 2021. 181, 105957.

27. Karl, J. W., & Sprinkle, J. E. Low-cost livestock global positioning system collar from commercial off-the-shelf parts. Rangeland Ecology & Management, 2019. 72(6), 954-958.

28. McGranahan, D. A., Geaumont, B., & Spiess, J. W. Assessment of a livestock GPS collar based on an open‐source datalogger informs best practices for logging intensity. Ecology and evolution, 2018. 8(11), 5649-5660.

29. Hampson, B. A., Morton, J. M., Mills, P. C., Trotter, M. G., Lamb, D. W., & Pollitt, C. C. Monitoring distances travelled by horses using GPS tracking collars. Australian Veterinary Journal, 2010. 88(5), 176-181.

- Lines 74-77. Authors must provide a bibliographic citation that accredits this sentence.

ANSWER: Two citations (29 and 30) are provided in the next sentence, in which we highlight that many authors found a strong relationship between the site use intensity gradient and the transition of specific vegetation communities. 

29. Peter M, Edwards PJ, Jeanneret P, Kampmann D, Lüscher A. Changes over three decades in the floristic composition of fertile permanent grasslands in the Swiss Alps. Agric Ecosyst Environ 2008; 125(1–4):204–12. 

30. Tasser E, Tappeiner U. Impact of land use changes on mountain vegetation. Appl Veg Sci. 2002;5(2):173–84.

- Lines 103-104. Authors should detail the types of buildings considered.

ANSWER: In the introduction, we specified as examples the farm centers and cowshed, whereas in the Material and Methods (i.e. ‘Computation of site-use intensity proxies’ section) we also indicated ‘other buildings related to pastoral activities’ which are buildings useful as a foothold for pastoral equipment (e.g. mobile electric fences, posts, energizers).

- Lines 120-121. This sentence requires a citation too.

ANSWER: We did not add a citation as in the previous paragraph (from L73 to L86) we discussed, with several references, about the relations between livestock site-use intensity and botanical composition.

- Line 182. Clarify if the distance considered was in a straight line or following the available paths.

ANSWER: We added a sentence to better explain the concept that: “Proximity was based on a straight-line distance between vegetation surveys and buildings.”

- Line 187. Clarify if for the calculation of the stocking rate only the animal that carried the GPS or the entire herd was considered.

ANSWER: Thank you for your comment. Actually, in this analysis we did not aim at calculating the stocking rate, but rather at assessing if slope and distance from buildings were good predictors of site-use intensity, measured by the number of GPS locations. As the term ‘stocking rate’ may be misleading in this context, we removed it from the sentence.

---

## [Decision Letter · Decision Letter 1]

13 Oct 2021

Slope and distance from buildings are easy-to-retrieve proxies for estimating livestock site-use intensity in alpine summer pastures

PONE-D-21-19475R1

Dear Dr. Pittarello,

We’re pleased to inform you that your manuscript has been judged scientifically suitable for publication and will be formally accepted for publication once it meets all outstanding technical requirements.

Kind regards,

Emmanuel Serrano, PhD

Academic Editor

PLOS ONE

Additional Editor Comments (optional):

Congratulations!

Reviewers' comments:

Reviewer's Responses to Questions

**Comments to the Author**

1. If the authors have adequately addressed your comments raised in a previous round of review and you feel that this manuscript is now acceptable for publication, you may indicate that here to bypass the “Comments to the Author” section, enter your conflict of interest statement in the “Confidential to Editor” section, and submit your "Accept" recommendation.

Reviewer #1: All comments have been addressed

2. Is the manuscript technically sound, and do the data support the conclusions?

Reviewer #1: Yes

3. Has the statistical analysis been performed appropriately and rigorously? 

Reviewer #1: I Don't Know

4. Have the authors made all data underlying the findings in their manuscript fully available?

Reviewer #1: Yes

5. Is the manuscript presented in an intelligible fashion and written in standard English?

Reviewer #1: Yes

6. Review Comments to the Author

Reviewer #1: (No Response)

7. PLOS authors have the option to publish the peer review history of their article (what does this mean?). If published, this will include your full peer review and any attached files.

Reviewer #1: **Yes: **Jordi Bartolomé Filella

---

## [Editor Report · Acceptance letter]

25 Oct 2021

PONE-D-21-19475R1 

Slope and distance from buildings are easy-to-retrieve proxies for estimating livestock site-use intensity in alpine summer pastures 

Dear Dr. Pittarello:

I'm pleased to inform you that your manuscript has been deemed suitable for publication in PLOS ONE. Congratulations! Your manuscript is now with our production department. 

Kind regards, 

on behalf of

Dr. Emmanuel Serrano 

Academic Editor

PLOS ONE